# Multi-Year Crop Type Mapping Using Sentinel-2 Imagery and Deep Semantic Segmentation Algorithm in the Hetao Irrigation District in China

Guang Li [1], Wenting Han [1,2,*], Yuxin Dong [1], Xuedong Zhai [1], Shenjin Huang [3], Weitong Ma [4], Xin Cui [1] and Yi Wang [5]

1 College of Mechanical and Electronic Engineering, Northwest A&F University, Yangling 712100, China
2 Key Laboratory of Agricultural Internet of Things, Ministry of Agriculture, Yangling 712100, China
3 Computer Science and Technology, Harbin Institute of Technology, Harbin 150001, China
4 Computer College of Water Resources and Architectural Engineering, Northwest A&F University, Yangling 712100, China
5 College of Information, Xi'an University of Finance and Economics, Xi'an 710100, China
* Correspondence: hwt@nwafu.edu.cn

**Abstract:** Accurately obtaining the multi-year spatial distribution information of crops combined with the corresponding agricultural production data is of great significance to the optimal management of agricultural production in the future. However, there are still some problems, such as low generality of crop type mapping models and susceptibility to cloud pollution in large-area crop mapping. Here, the models were constructed by using multi-phase images at the key periods to improve model generality. Multi-phase images in key periods masked each other to obtain large-area cloud-free images, which were combined with the general models to map large areas. The key periods were determined by calculating the global separation index (GSI) of the main crops (wheat, maize, sunflower, and squash) in different growth stages in the Hetao Irrigation District (HID) in China. The multi-phase images in the key period were used to make the data set and were then combined with a variety of deep learning algorithms (U-Net, U-Net++, Deeplabv3+, and SegFormer) to construct general models. The selection of the key periods, the acquisition of regional cloud-free images, and the construction of the general crop mapping models were all based on 2021 data. Relevant models and methods were respectively applied to crop mapping of the HID from 2017 to 2020 to study the generality of mapping methods. The results show that the images obtained by combining multi-phase images in the key period effectively avoided the influence of clouds and aerosols in large areas. Compared with the other three algorithms, U-Net had better mapping results. The F1-score, mean intersection-over-union, and overall accuracy were 78.13%, 75.39% and 96.28%, respectively. The crop mapping model was applied to images in 2020, and its average overall accuracy was more than 88.28%. When we applied the model to map crops (county food crops, cash crops, and cultivated land area) from 2017 to 2019, the regression analysis between the mapping areas obtained by the model and the ground measurements was made. The R2 was 0.856, and the RMSE was 17,221 ha, which reached the application accuracy, indicating that the mapping method has certain universality for mapping in different years.

**Keywords:** deep semantic segmentation; Sentinel-2 imagery; multi-year crop mapping; U-Net; global separability index

## 1. Introduction

With the rapid growth of population and urbanization, the distribution pattern of crops has gradually changed [1,2]. Rapid and accurate information on the multi-year spatial distribution and spatial and temporal dynamics of crops in planting areas is a hot research topic [3–5]. Multi-year crop distribution information is the data basis for monitoring crop

planting area, predicting regional crop yield, ensuring regional food security balance, and optimizing planting structure [6–8]. Compared with manual survey, remote sensing technology is a more accurate and intuitive method to monitor crop distribution, and has been widely used in the extraction of large-area crop distribution information. [9,10].

Over the last two decades, with the development of remote sensing technology, many Earth observation satellites have emerged [11]. Satellites with medium and high resolution, such as Sentinel-2A/B, Landsat 8, and Gaofen (GF), are widely used in crop mapping in the world [12–14]. China is an agricultural and populous country. Its agricultural production mostly adopts the family responsibility contract system, resulting in the fragmentation of its cultivated land (e.g., the average field size is less than 1 ha). Furthermore, in the same growing season, different crops may be planted in different fragmented cultivated land, which makes the planting situation in the planting area more complex. This puts forward higher requirements for remote sensing monitoring technology [15] in terms of spatial resolution, spectral resolution, temporal resolution, and data cost. Sentinel-2 series satellites, with a spatial resolution of 10 m and a revisit period of 5 days, can obtain 13 bands, making them an important monitoring means for crop mapping in complex farming areas [13,16,17].

The research on farmland crop mapping using optical remote sensing includes the following two aspects. The first is crop mapping methods based on time series. Agricultural land system is a spatial expression formed by the planting and combination of a variety of crops through specific laws. Due to the different growth and development of crops, a single remote sensing image cannot cover all growth and development stages of crops. Therefore, crop type mapping using phenological change law is the most common method. Time series remote sensing image mapping is mainly studied from the selection of the vegetation index and the construction of a high time resolution curve and mapping method [18,19]. More than 40 vegetation indices have been defined and successfully applied to all aspects of agricultural production, among which NDVI and EVI are the most commonly used indexes [20]. The construction of high time resolution curves is mainly through the selection of satellite images with high time resolution and the collaborative construction of a variety of satellite images [14]. The commonly used mapping methods include the threshold method, dynamic time warping technique (TWDTW), and machine learning method [21,22]. When time series images are used to map large areas, medium and low-resolution remote sensing images are often used, which leads to low mapping accuracy. This method is often suitable for crop mapping with large spatial scale and obvious seasonal rhythm. This strategy, however, necessitates the collection of all data throughout the whole development phase, which limits its usefulness. The second aspect is crop mapping through key period images. Most studies focus on the construction and screening of classification features, as well as classification algorithms [23–26]. The construction and screening of crop mapping features is mainly based on the color, texture, spectral band, and other aspects of the image, and is combined with the methods of feature weight and separability [27,28]. Most of these features are extracted from single period images, and their applicability in different time is not clear. The classification algorithms mainly focus on machine learning algorithms and deep learning algorithms [29,30]. Compared with a machine learning algorithm, a deep learning algorithm has the property of convolution and can consider the characteristics of neighboring pixels, which has been proved to have good cartographic ability [31–33]. Recurrent neural networks (RNN), convolutional neural networks (CNN), and full convolutional neural networks (FCN) are the most widely utilized deep learning networks. The FCN is more effective than CNN and RNN, and frequently maps crop types through semantic segmentation. [34]. Crop mapping, which falls under the field of semantic segmentation, primarily classifies each pixel in the image with a category. Since 2012, a number of semantic segmentation models have been developed [35], such as the U-Net series model [36,37], Deeplab series model [38], and Transformer model [39], which have achieved good application in image deep semantic segmentation. Most studies use semantic segmentation algorithms to build crop mapping models, which are basically

based on images from a single period or a few periods. As a result, the application of the model on a spatio-temporal scale has limitations [29]. At the temporal scale, whether the deep semantic segmentation algorithm is suitable for the construction of crop mapping models from multi-phase remote sensing images needs further study. Meanwhile, the application of the constructed model in different years also needs to be further studied. At the spatial scale, whether the model is suitable for high-quality crop mapping in large areas needs further study [34–36].

Therefore, the problems need to be solved for accurate multi-year crop type mapping over a large area. Firstly, the applicability of features, algorithms, and constructed models in different time scales should be clarified. Secondly, it is necessary to explore the ability of the model to map crops over large areas. The image of crop mapping in large areas is susceptible to cloud pollution and missing data [14,40]. Based on the aforementioned context, we utilized the Sentinel-2 data from May to September 2020 and 2021 for the Inner Mongolia Autonomous Region of China's Hetao Irrigation District (HID) as an example. We explored the efficient general mapping methods for large areas of the four crops (wheat, maize, sunflower, and squash), and applied these methods to map the four crops in the HID from 2017 to 2019. Meanwhile, this paper answers the following three questions in the research process: (1) What are the advantages of the constructed mapping models for crop type mapping in large areas? (2) What is the most suitable algorithm for mapping the four crops in the HID? (3) What is the generalization of the proposed mapping method in multi-year crop mapping?

## 2. Materials and Methods

### 2.1. Study Area

Hetao Irrigation District (HID) is the third largest irrigation district in China and is located in Bayannur City in the Inner Mongolia Autonomous Region. With 570,000 hectares of irrigated farmland, it is a significant grain production region. The research region is shown in Figure 1 and comprised of five counties. The region is bright, sunny, dry, windy, and has only moderate amounts of rainfall (annual average rainfall 170 mm). The Yellow River provides most of the irrigation water, and the altitude of the HID is between 1000 and 1091 m above sea level.

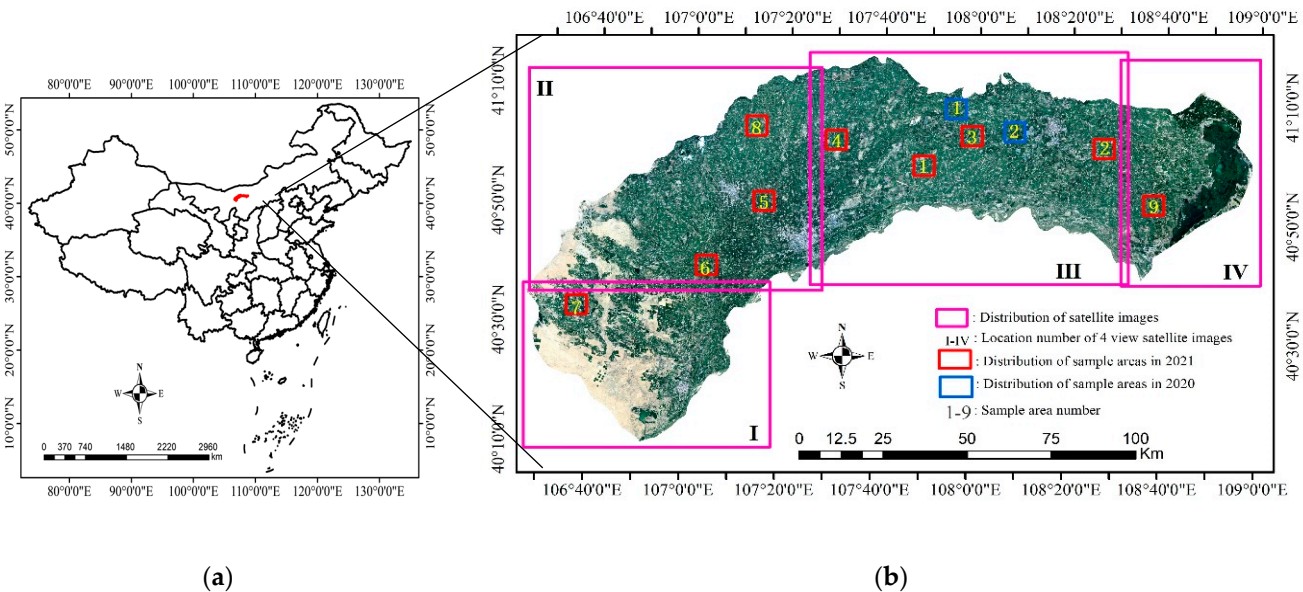

(**a**)                                                              (**b**)

**Figure 1.** Aerial map of the study area. (**a**) Location of Hetao Irrigation District, Inner Mongolia, China; (**b**) the distribution of sampling areas in Hetao Irrigation District in 2020 and 2021.

Summer maize, sunflower, spring wheat, and squash are the main crops in the study region, according to local government and field research. Crops grow between the months of April and October. The four crops' planting areas are determined using local government statistics, which are available at http://tjj.bynr.gov.cn/ (accessed on 12 April 2021). Between 2015 and 2020, the amount of maize and sunflower planted in the HID essentially constituted 35% to 40% of the total cultivated land area of the irrigation area, while wheat accounted for 10%, and squash and other cash crops accounted for roughly 10%. Therefore, the study that comes next concentrates on the planting of wheat, maize, sunflowers, and squash. The blue box in Figure 1b shows two sampling areas in 2020, and the red box in Figure 1b shows the location distribution of nine sampling areas in 2021.

The phenological calendar of four main crops in HID is shown in Figure 2. The Sentinel-2 images that correspond to the stages of crop growth in 2020 are shown by the black vertical lines, and the Sentinel-2 images that correspond to the stages of crop growth in 2021 are indicated by the red vertical lines. The growth period of spring wheat is from April to July, that of summer maize and squash is from May to October, and that of sunflower is from June to October. Because the phenological calendars of different crops in HID are different, using images in different periods to map different crops is needed.

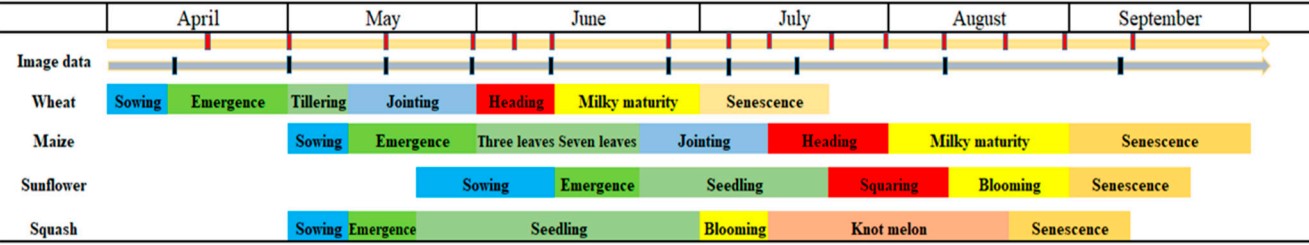

**Figure 2.** The phenology calendar of the four crops.

### 2.2. Image Capture and Processing for Sentinel-2

Sentinel-2 images with bidirectional enhancement were used. Available images of the study area for 2017–2021 were downloaded from the USGS Earth probe data portal (https://earthexplorer.usgs.gov/ accessed on 1 January 2022). To reduce the impact of weather factors on the data, all available data were filtered with a cloud content of no more than 20%, and the final data were shown in Table 1. In 2020, all available images from April to September were obtained in region III, which were mainly used to test the applicability of the model in different years. In 2021, 15 available images of regions I–IV from April to September were obtained, which were mainly used to train crop mapping models. The images obtained in 2017–2019 were mainly used for the application of the method in crop mapping in large areas over multiple years. By combining the images of different areas in the key period, we obtained the high-quality images of large areas for high precision mapping. Section 2.3.2 described in detail the methods for obtaining high quality images of large areas. For example, in 2019, the high-quality images of region I, region II, region III, and region IV were obtained on 30 May, 1 Jun, 22 May, and 27 May, respectively. These times are all in the key period of mapping. The images of four areas in different times were spliced to obtain large area high-quality images, which could avoid cloud pollution.

**Table 1.** The Sentinel-2 imagery used in the study.

| Year | Distribution of Satellite Images | | | |
|---|---|---|---|---|
| | I * | II * | III * | IV * |
| 2021 | Apr 16, May 1, May 16, May 31, Jun 5, Jun 10, Jun 25, Jul 5, Jul 10, Jul 20, Jul 30, Aug 9, Aug 19, Aug 29, Sept 8 | | | |
| 2020 | May 31, Sept 18 | May 31, Sept 18 | Apr 11, May 1, May 16, May 31, Jun 10, Jun 25, Jul 5, Jul 15, Aug 9, Sept 3 | May 31, Sept 18 |
| 2019 | May 30, Aug 18, Sept 2 | Jun 1, Aug 18, Sept 2 | May 22, Aug 15, Sept 4 | May 27, Aug 15, Sept 4 |
| 2018 | May 22, Aug 5, Sept 9 | May 22, Aug 5, Sept 9 | May 22, Aug 5, Sept 9 | May 22, Aug 5, Sept 9 |
| 2017 | May 30, Aug 5, Sept 7 | May 30, Aug 5, Sept 27 | May 27, Aug 5, Sept 24 | May 27, Aug 5, Sept 24 |

* I, II, III, and IV are the location numbers of the 4-scene satellite images monitoring the HID, as shown in the purple box in Figure 1b.

Multi-period and multi-scene images were used in the study, so it is important to control image quality. We started with the Sentinel-2 Multispectral Instrument (MSI) Level-1C data set. The Level-1C bands represent the top of atmosphere (TOA) reflectance without atmospheric correction. The atmospheric conditions (warm at ground level and cold at altitude) in the semi-arid region will lead to continuous haze and water vapor in the daytime under the absence of wind, which will affect the image quality. To protect the image from atmospheric reflection, the Sen2SOR tool was used to correct the image and to generate Level-2A images. Table 2 compared the mean digital number (DN) values and standard deviations of the Level-1C and Level-2A images of residential areas (622 pixels) on May 31 and June 10, 2020. After atmospheric correction, the DN value and standard deviation of the Level-2A images are both larger, indicating that haze and water vapor in the atmosphere have an impact on image quality. Therefore, Sen2COR-2.5.0 was used in this paper to convert all Level-1C images into Level-2A images for subsequent research. The SNAP6.0 (Sentinel Application Platform) resampled the L2 level images. The WGS 1984 coordinate system and UTM48N projection are used for all images. Given that four scenes are necessary to cover the full research region, each band was seamlessly combined in ENVI5.3 after resampling to provide a picture of the entire study area. Additionally, the pictures were cropped in ENVI5.3 to obtain the outcomes seen in Figure 1b.

**Table 2.** Reflection of Sentinel-2 spectral bands at Level-1C and Level-2A.

| Bands | Level-1C | | | | Level-2A | | | |
|---|---|---|---|---|---|---|---|---|
| | 30 May | | 10 Jun | | 30 May | | 10 Jun | |
| | Mean * | Std * | Mean * | Std * | Mean * | Std * | Mean * | Std * |
| B1 | 1751 | 165 | 1759 | 146 | 1139 | 275 | 1146 | 269 |
| B2 | 1897 | 327 | 1849 | 298 | 1642 | 491 | 1678 | 502 |
| B3 | 1888 | 363 | 1981 | 337 | 2093 | 511 | 2144 | 522 |
| B4 | 2344 | 430 | 2409 | 403 | 2639 | 530 | 2681 | 537 |
| B5 | 2541 | 362 | 2517 | 331 | 2781 | 452 | 2805 | 443 |
| B6 | 2801 | 277 | 2845 | 256 | 3014 | 326 | 3053 | 318 |
| B7 | 3065 | 268 | 3074 | 244 | 3214 | 302 | 3220 | 290 |
| B8 | 3052 | 244 | 3063 | 229 | 3422 | 291 | 3437 | 286 |
| B8A | 3279 | 259 | 3239 | 236 | 3342 | 290 | 3388 | 280 |
| B9 | 1550 | 109 | 1659 | 144 | 3279 | 228 | 3304 | 214 |
| B11 | 3839 | 493 | 3772 | 453 | 4164 | 544 | 4130 | 510 |
| B12 | 3220 | 438 | 3180 | 422 | 3655 | 501 | 3607 | 484 |

* The mean digital number (DN) values and standard deviations of the Level-1C and Level-2A images of residential areas (622 pixels) on 31 May and 10 June 2020.

### 2.3. Methods

Data preparation, feature screening, and data set production, classification, and comparative analysis are the three components of the research (Figure 3). First, we preprocessed the downloaded Sentinel-2 images with radiometric correction, resampling, splicing, and cutting, and gathered and organized the ground survey data. Second, establish the critical period and features for mapping by calculating the global separation index of crops in 12 bands in different periods. Make the training set, validation set, and test set of the model. Finally, the mapping models are obtained by combining U-Net, U-Net++, Deeplabv3+, and SegFormer with the preferred bands. The applicable model is obtained by evaluating the accuracy of the 2021 test area image combined with a confusion matrix. The obtained model is applied to the mapping of the test area in 2020, and the applicability of the model in different years is evaluated combined with the confusion matrix. The method is applied to large area crop mapping in 2017–2021 and compared with the statistical data to determine the applicability.

#### 2.3.1. The Global Separability Index Creation

The construction of the global separability index (GSI) involved two steps. Firstly, the separability ($SI_{so}$) of a particular crop from other crops was calculated based on intraspecific variability and interspecific variability [41]. A set of values characterized the separability of a particular crop from other crops in pairs. Secondly, in order to represent the identifiability of crops clearly, a set of separable values corresponding to specific crops were averaged and denoted as the GSI. It could describe the capacity of the model to recognize various crops at various times and under various bands. The following are the formulas used to determine the $SI_{cj}$, and GSI:

$$SI_{so}(j,k) = \frac{\Delta inter(s,o)}{\Delta intra(s,o)} = \frac{|\overline{u_s} - \overline{u_o}|}{1.96 \times (\sigma_s + \sigma_o)}$$

where

$$j = \{B_1, B_2, B_3, B_4, B_5, B_6, B_7, B_8, B_{8a}, B_9, B_{11}B_{12}\},\ k \tag{1}$$

$$GSI(j,k) = Average\left(\sum_{o=1}^{z} SI_{so}\right)$$

where

$$j = \{B_1, B_2, B_3, B_4, B_5, B_6, B_7, B_8, B_{8a}, B_9, B_{11}B_{12}\},\ k \tag{2}$$

where s stands for a particular crop and o for additional, undesignated crops; j stands for different bands and k stands for different periods; $\overline{u_s}$ and $\overline{u_o}$ represent the mean spectral values of band value j on data k for the particular crop and endmember class o, respectively; $\sigma_s$ and $\sigma_o$ represent the standard deviations of that feature within the particular crop and class o; z is the total number of a class' o.

#### 2.3.2. Data Set Construction Method

The construction of the data set includes the selection of sample areas and field investigation, the production of crop type labels in sample areas, and the division of the data set. Firstly, the remote sensing images are visually discriminated in the early growth stage, and the sampling areas are selected according to the number of ground objects and distribution ratio. In 2020 and 2021, two and nine sample areas were selected respectively, as shown in Figure 1b,c. The mobile phone software GPS toolbox (5 m precision) was used to record the types of crop and geographical coordinates of the sample areas in July–August 2020 and June–July 2021, respectively. Secondly, in ArcGIS (10.2), the ground record data was combined with the visual discrimination method to sketch the vector map of crop distribution on the remote sensing image layer and convert it into a raster map as the true value map of crop distribution in the sample area. Because crops have distinct key growth stages, it is vital to identify crops using images at various times to determine crop

distribution precisely. A separate true distribution map for each crop is needed, so the true distribution maps are produced for the major crops in 11 sampling areas (SA), measuring 6 km × 6 km, in 2020 and 2021.

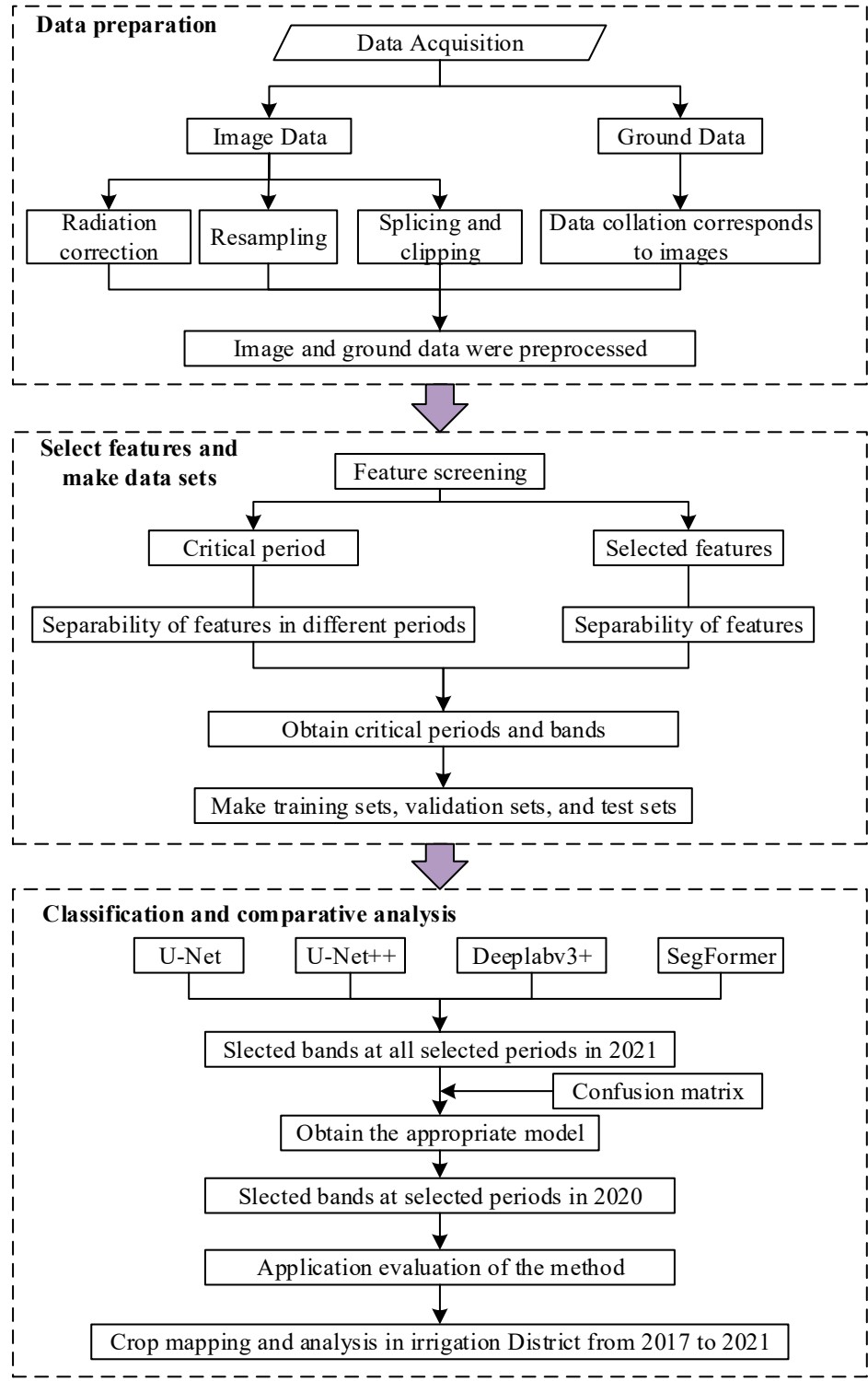

**Figure 3.** Research flowchart.

In 2021, the multi-phase original images used for modeling in sampling area 2 during the key period for mapping and the labels containing maize and non-maize were shown in Figure 4a,b. The data and labels of other crops and regions are similar, but they are

not shown in full due to length. The percentages of each crop in the sample areas are presented in Table 3. In this study, SAs 1, 2, 3, 4, 5, 6, and 7 in 2021 were chosen as training-validation areas, while SAs 8 and 9 in 2021 were chosen as test areas. To evaluate the model's applicability to other years, SAs 1 and 2 in 2020 were employed. Direct training would cause the graphics processing unit's (GPU) memory to become exhausted due to the enormous picture size of a sample region. The original picture (Figure 4a) and labeled image (Figure 4b) must be cut into small images (Figure 4c,d) with a resolution of 128 × 128 pixels in order to circumvent the GPU's memory restriction. The training set and validation set were selected in the proportion of 3:1. The data set would be constructed after the key period was determined, so the results of data set construction will be described in detail in the Section 3.

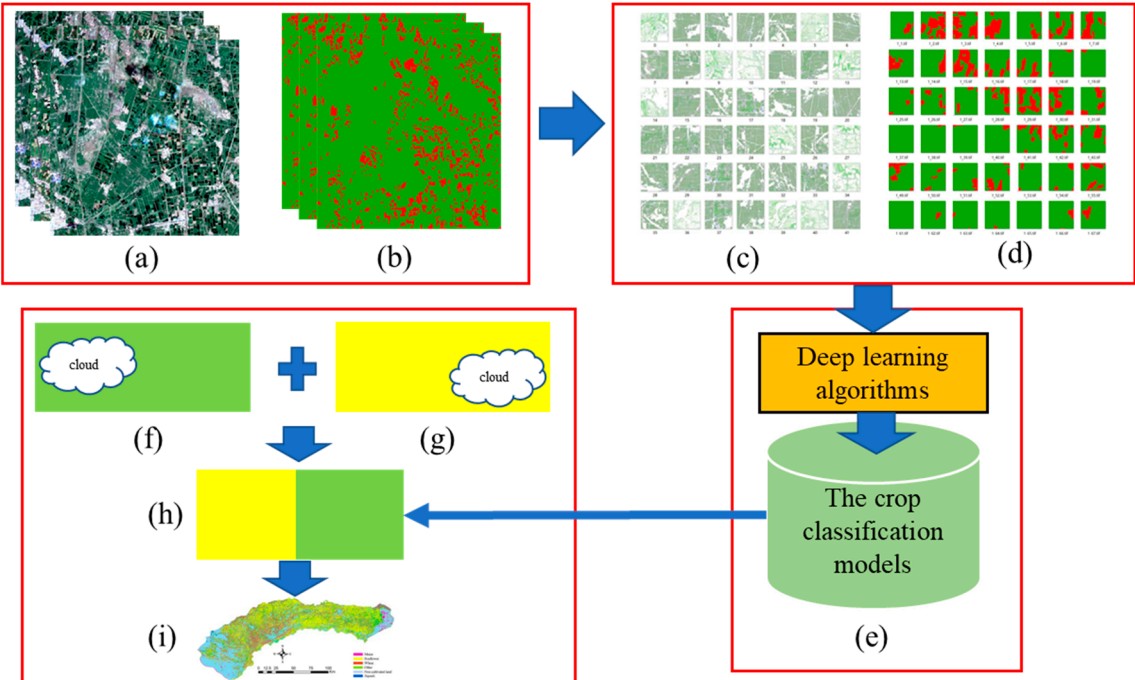

**Figure 4.** Diagram of data set construction and classification process. (**a**) Multi-phase raw images of SA 2 for maize recognition. (**b**) The maize labels of SA 2. (**c**) The cropped original images. (**d**) The clipped labels. (**e**) Schematic diagram of classification model acquisition. (**f**,**g**) Diagram of cloud influence in the study area at different times. (**h**) Schematic diagram of high-quality image acquisition in the study area. (**i**) Crop mapping results from the study area.

**Table 3.** The percentage of lands used for crop planting in each sample area.

| Year | SA | Wheat (%) | Maize (%) | Sunflower (%) | Squash (%) | Others (%) |
|---|---|---|---|---|---|---|
| 2021 | 1 | 0.37 | 10.49 | 45.53 | 4.74 | 38.86 |
|  | 2 | 0.67 | 8.51 | 56.69 | 5.54 | 28.59 |
|  | 3 | 1.13 | 26.61 | 40.22 | 7.78 | 24.27 |
|  | 4 | 1.27 | 22.41 | 38.99 | 6.32 | 31.00 |
|  | 5 | 4.23 | 40.43 | 26.24 | 1.84 | 27.26 |
|  | 6 | 1.41 | 28.39 | 27.67 | 1.94 | 40.58 |
|  | 7 | 2.41 | 16.62 | 26.71 | 6.61 | 47.65 |
|  | 8 | 1.47 | 13.12 | 46.34 | 8.03 | 31.04 |
|  | 9 | 0.39 | 24.94 | 45.01 | 4.45 | 25.21 |
| 2020 | 1 | 3.23 | 11.00 | 31.88 | 3.37 | 50.52 |
|  | 2 | 2.99 | 20.57 | 33.98 | 1.03 | 41.42 |

In addition to training and validation sets, test sets are also needed to evaluate the applicability of the model. The test set can be the images of the sample area excluding the construction training and validation sets, or the images of the whole research area, which are easily affected by clouds, as shown in Figure 4f,g. The classification model obtained by multi-phase images in the key period (Figure 4e) is applicable to map each image in the key period. To eliminate the influence of cloud on crop mapping in a large area, multi-phase images were combined in a key period by mask. The specific combination process of multi-phase images was as follows. Firstly, images at different times in key time periods were obtained, as shown in Figure 4f,g. Secondly, the location of the cloud on the image in different periods was determined and the area was masked. Finally, the Seamless Mosaic tool in ENVI was used to mosaic the multi-phase images after the mask, so as to obtain the high-quality images (Figure 4h) of the study area as the test input of the mapping model.

### 2.3.3. U-Net Classification Algorithm

The U-Net algorithm was initially suggested by Ronneberger et al. (2015) to segment biomedical images [36]. According to the processing method in reference [42], after the fourth and fifth convolution processes, a dropout layer with a probability of 0.5 was added to prevent overfitting. During each training iteration, neurons were discarded with a rate of 0.5. After each convolutional procedure, a BN (batch normalization) process was implemented to increase the network training efficiency. Figure 5 depicted the U-Net structure. The early aspects of the contraction route, which include rich but abstract spatial information, are represented by the white box. The up-sample convolution result is shown by the yellow box, which contains detailed features with little spatial information that were extracted from the whole architecture. The D stands for the dropout processing outcome.

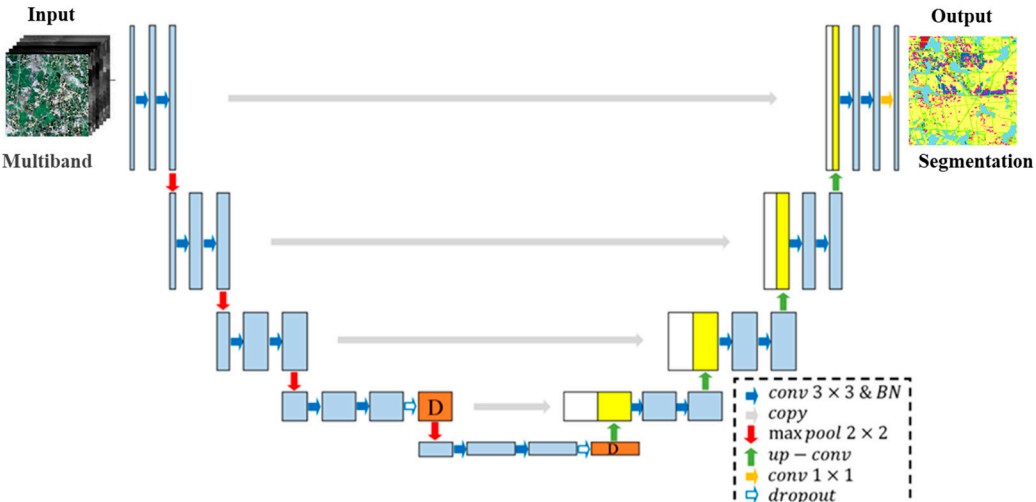

**Figure 5.** Illustration of the U-Net structure.

### 2.3.4. U-Net++ Classification Algorithm

The NET++ algorithm is an improvement in jump connection made by Zhou et al. (2018) on the basis of U-Net. Compared with the original 4-layer U-Net, U-Net++ connects all the U-shaped structures of the remaining 1~3 layers together. This complementary method of long connection and short connection makes the semantic feature level of adjacent levels of the encoder closer. When the received encoder feature map is more semantically similar to the corresponding decoder feature map, the network optimization ability will be greatly improved. The structure of U-Net++ is shown in the literature [37]. The network is composed of U-Net with different depths in 1–4 layers. Different network layers form a complete U-shaped network by fusing the features sampled on the decoder with the module features corresponding to the next layer.

### 2.3.5. Deeplabv3+ Classification Algorithm

The fourth-generation Deeplabv3+ product was enhanced by Chen et al. (2018) for the Deeplab series network. The structure described in the literature [38]. An encoder-decoder design is used to combine information from different scales. The largest advancement is the ability to enlarge the receptive field and have each convolution output a larger range of information without sacrificing information by combining depth separable convolution with cavity convolution. In place of all max pooling procedures, deep separable convolution is used for the ASPP and decoder, which can significantly lower the number of parameters and speed up model training.

### 2.3.6. SegFormer Classification Algorithm

SegFormer is a simple and efficient semantic segmentation framework proposed by Xie et al. (2021) by combining transformer architecture with a lightweight multi-layer perceptron (MLP) decoder. Its structure shown in the literature [39]. The encoder and decoder are the two modules that make up SegFormer. To obtain multi-level features, the image is segmented into $4 \times 4$ small blocks and input into the hierarchical transformer encoder. In order to forecast the segmentation mask at $H/4 \times W/4 \times N_{cls}$ resolution, these multi-level features are then sent to the MLP decoder, where H, W, and Ncls are the height, width, and number of categories of the image, respectively. The encoder includes four transformer blocks, and the layered feature with a resolution of $H/2^{i+1} \times W/2^{i+1} \times C_i$ is obtained after each transformer block. Each transformer block consists of efficient self-attention, Mix-FFN, and overlap patch merging. Among them, effective self-attention is used to improve the computational efficiency. Mix-FFN is used to solve the low accuracy problem caused by the inconsistency between the image resolution and the training image resolution. Overlap patch merging is used to maintain the local continuity around these patches. The decoder includes four steps. Firstly, the multi-level characteristics of the encoder unify the channel size through an MLP layer. Then, the features are sampled up to 1/4 and connected together. Thirdly, MLP layer fusion splicing feature is adopted. Finally, another MLP layer uses the fusion feature to predict the image.

### 2.3.7. Hyperparameter Settings

Python 3.8.5, Anaconda 3, and PaddlePaddle 2.3 are used as the programming language, development environment and deep learning framework, respectively. Experiments are conducted on high-performance computing resources, such as operating system (Windows 10 Enterprise 64 bit), CPU (Intel Core i9-10920X CPU @ 3.50 GHz), and accelerator (NVIDIA RTX3090 GPU with 24 GB RAM)). According to the performance of computing resources and the number of samples of experimental data, the batch size and iteration times are set to 24 and 20,000, respectively. The stochastic gradient descent (SGD) algorithm was used for training optimization, and the momentum parameter and weight attenuation parameter was set to 0.9 and $4.0 \times 10^{-5}$, respectively. To ensure the smooth progress of the training process, the multi learning rate strategy is adopted. Initial_learning_rate and the power parameter were set to $10^{-3}$ and 0.9, respectively.

### 2.3.8. Precision Evaluation

The terms $F_1$-score, overall accuracy (OA), and mean intersection-over-union (mIoU), which may comprehensively describe the accuracy of pixel types and object locations in diverse regions, are frequently used to assess the accuracy of mapping [43,44]. The following are the formulas used to determine the OA, F1-score, and mIoU:

$$OA = \sum_{c=1}^{n} \frac{TP_c + TN_c}{TP_c + TN_c + FP_c + FN_c} \tag{3}$$

$$precision_c = \frac{TP_c}{TP_c + FP_c} \tag{4}$$

$$recall_c = \frac{TP_c}{TP_c + FN_c} \tag{5}$$

$$F1\text{-}scroe = \frac{2 \times precision \times recall}{precision + recall} \tag{6}$$

$$IoU = \frac{TP_c}{TP_c + FP_c + FN_c} \tag{7}$$

$$mIoU = \frac{\sum_{c=1}^{n} IoU}{n} \tag{8}$$

where $n$ is the number of categories, and $c$ denotes a particular category; *TP* stands for the proper classification's beneficial positive classes; *FP* stands for the inaccurate classification's negative classes; *TN* indicates that an observation is predicted to be negative and is actually negative; *FN* indicates that observation is predicted to be negative and is actually positive. *Recall*, which stands for sensitivity, is calculated as the proportion of test data set positive samples to the total number of positive samples. *Precision* is the fraction of all precise positive and negative classes for a given class throughout the whole sample.

## 3. Results

### 3.1. Feature Screening and Data Set Construction Results

The GSI thermal maps of the four crops at various wavelengths and throughout various development phases are displayed in Figure 6. Each unit denotes the GSI value for a particular band at a particular time. The deeper the hue, the higher the value, and the stronger the band's categorization ability in this time period. There were variations in GSI for the four crops during the course of the research. For wheat, maize, squash, and sunflower, the highest GSI was higher than 1.6, 1.2, 1, and 0.7, respectively. Distinct crops also had different peak times and band characteristics in the GSI. For wheat, the corresponding bands B2-B5 and B11-B12 between May 15 and June 25 had higher GSI values. For maize, the corresponding bands B2-B5 and B11-B12 between July 30 and September 8 had higher GSI values. For sunflower, the corresponding bands B3-B4 and B6-B8A between July 30 and August 29 had higher GSI values. For squash, the corresponding bands B2-B5 and B11-B12 between August 9 and September 8 had higher GSI values. The GSI can characterize the key period of crop identification and the importance of each band in each time period. It can be used to determine the key period of classification and to remove redundant information. According to the value of the GSI and its distribution in the thermal map, the image and band selection results of the four crops in the key mapping period are determined (Table 4). Five images of wheat and maize were obtained in the key period of mapping, with the bands of B2-B5, and B11-B12. Four images of sunflower and squash were obtained in the key period of mapping, with the bands of B3-B4 and B6-B8A, and B2-B5 and B11-B12, respectively. The multi-stage images of each crop in the key period were cut, and finally 14,000, 14,000, 11,200, and 11,200 key period mapping data sets of wheat, maize, sunflower, and squash were obtained, respectively.

**Table 4.** Imagery and feature selection results of key period of mapping for the four crops. The results of making data sets using multi-stage images of the four crops in the key periods.

| Crops | Key Period | Selected Features | Data Sets |
|---|---|---|---|
| Wheat | May 16, May 31, Jun 5, Jun 10, Jun 25 | B2, B3, B4, B5, B11, B12 | 14,000 |
| Maize | Jul 30, Aug 9, Aug 19, Aug 29, Sept 8 | B2, B3, B4, B5, B11, B12 | 14,000 |
| Sunflower | Jul 30, Aug 9, Aug 19, Aug 29 | B3, B4, B6, B7, B8, B8A | 11,200 |
| Squash | Aug 9, Aug 19, Aug 29, Sept 8 | B2, B3, B4, B5, B11, B12 | 11,200 |

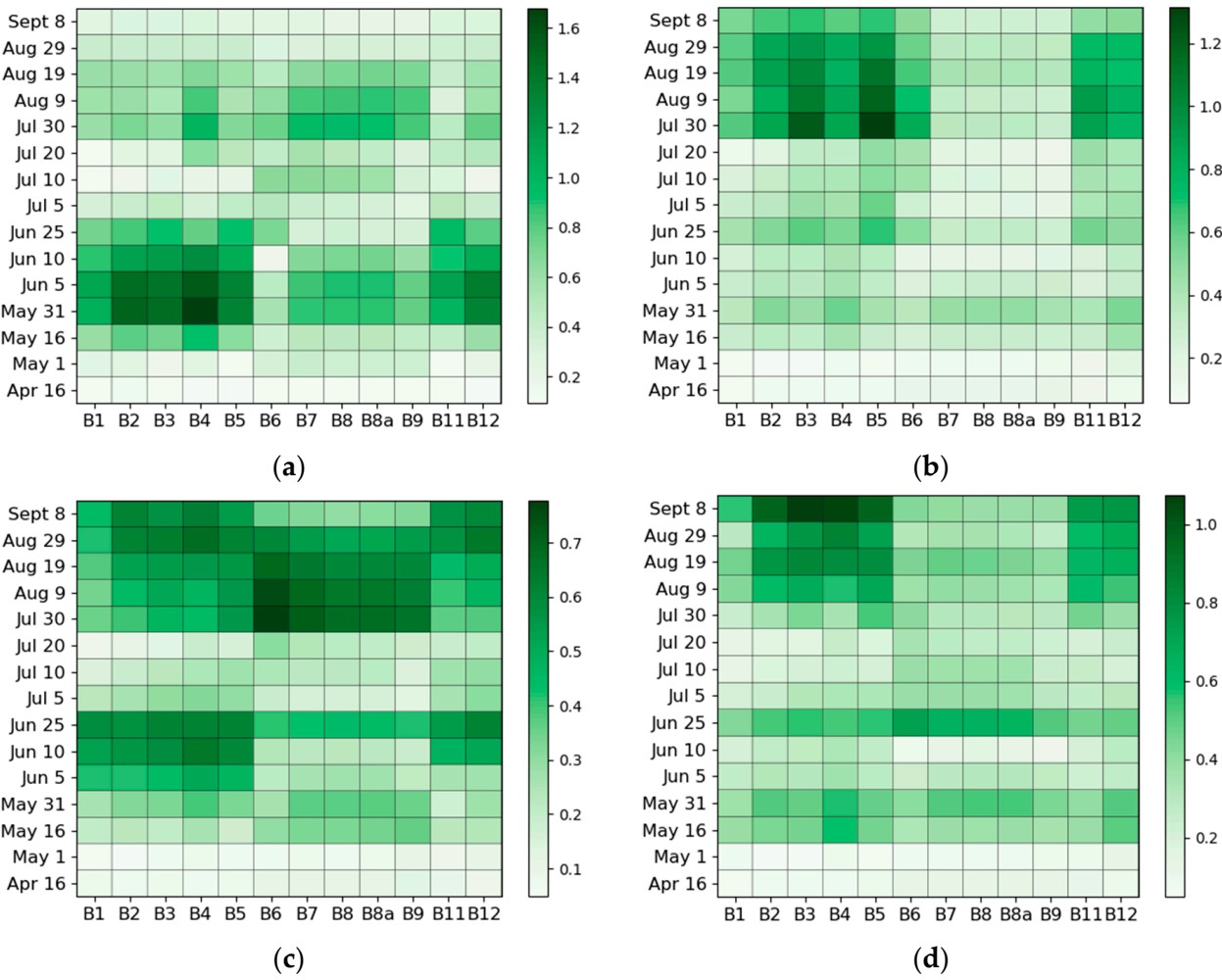

**Figure 6.** The GSI of various bands throughout various growth periods. (**a**) Wheat; (**b**) maize; (**c**) sunflower; (**d**) squash.

*3.2. Mapping Accuracy of Different Algorithms in 2021*

To determine the crop mapping method suitable for the HID in the key period, U-Net, U-Net++, Deeplabv3+, and SegFormer were used to construct the mapping model of four crops in the key period. Figure 7 shown the highest and lowest accuracy obtained by the four algorithms in corresponding periods. The mapping accuracy of U-Net for the four crops in the key period was better than the other three methods. The F1-score, mIoU, and OA can reach 69.16%, 73.91%, and 96.28%, respectively, for wheat, 72.68%, 74.78%, and 93.15%, respectively, for maize, 78.13%, 66.26%, and 80.79%, respectively, for sunflower, and 71.22%, 75.39%, and 95.71%, respectively, for squash. Meanwhile, for multi-stage image mapping in the key period, the four algorithms obtained the same period corresponding to the highest accuracy and the lowest accuracy. The highest and lowest mapping accuracy were obtained, respectively, on June 10 and May 16 for wheat, on July 30 and September 8 for maize, on August 29 and July 30 for sunflowers, on September 8 and August 9 for squash. The general mapping models have different mapping accuracy for images at different periods in the key period. In later application, priority should be given to selecting images with high mapping accuracy in the key period.

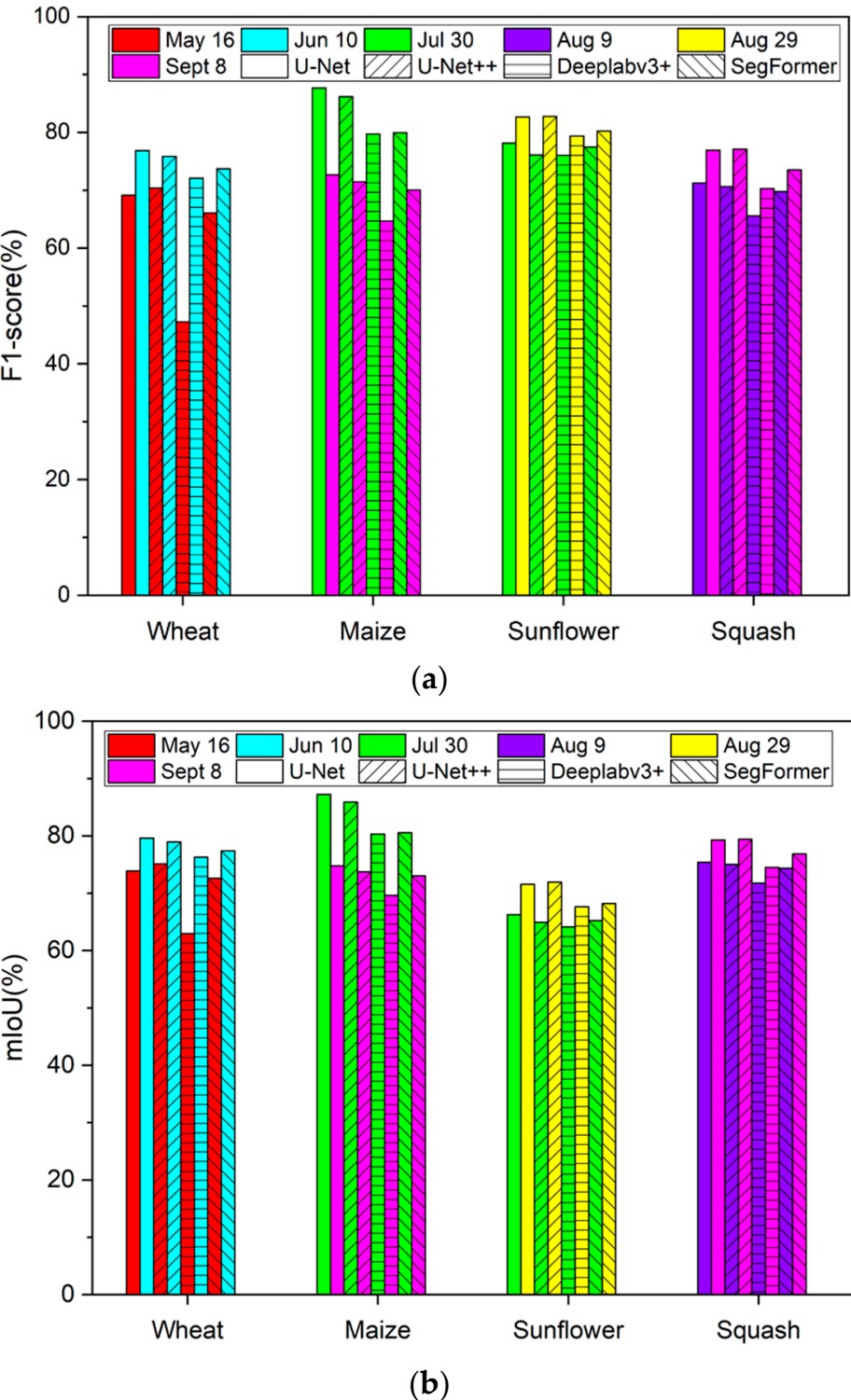

**Figure 7.** *Cont.*

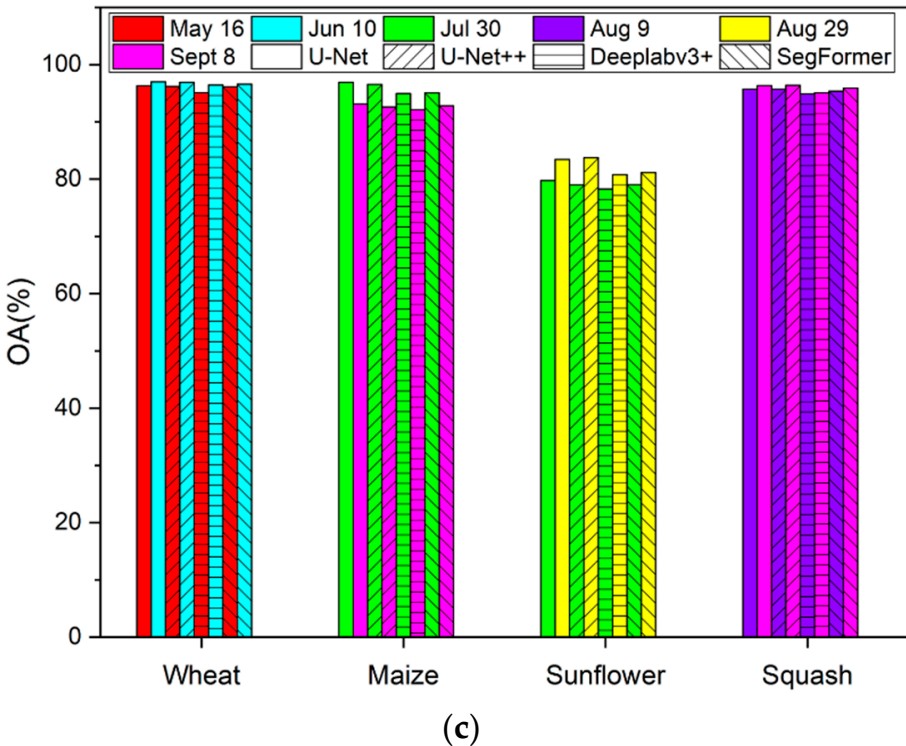

(**c**)

**Figure 7.** The highest and lowest results for the mapping of crops using selected features during key periods in 2021. (**a**) F1-score, (**b**) mIoU, and (**c**) OA for different algorithms. Different colors represent different growth periods. The histograms without slashes represent the U-Net algorithm. The histograms with slashes from the right to the left represent the U-Net++ algorithm, while those with slashes from the left to the right represent the SegFormer algorithm. The histograms with horizontal lines represent the Deeplabv3+ algorithm.

### 3.3. Mapping Results of the Model in 2020 and 2017–2019

Sections 3.1 and 3.2 determine the key periods of the four crops mapped in the HID and construct a general mapping model for the key periods of mapping through U-Net, respectively. Two sample areas in 2020 were used to study the applicability of general mapping methods in different years. During the key mapping period of four crops, two images were selected. Wheat (May 16, Jun 25), maize (Aug 9, Sept 3), sunflower (Aug 9, Sept 3), and squash (Aug 9, Sept 3) were mapped with general models, respectively. In Table 5, the F1-score, mIoU, and OA can be seen to reach 68.26%, 64.35%, and 96.93%, respectively, for wheat, 68.92%, 66.48%, and 85.14%, respectively, for maize, 68.52%, 65.32%, and 74.74%, respectively, for sunflower, and 68.69%, 63.28%, and 96.99%, respectively, for squash.

**Table 5.** The mapping accuracy of the model for the four crops in 2020.

| Crops | Data | Area | F1-score | mIoU | OA | Crops | Data | Area | F1-score | mIoU | OA |
|---|---|---|---|---|---|---|---|---|---|---|---|
| Wheat | May 16 | 1 | 71.78 | 66.075 | 97.25 | Maize | Aug 9 | 1 | 73.69 | 75.615 | 93.53 |
| | | 2 | 69.78 | 69.96 | 97.31 | | | 2 | 83.8 | 81.63 | 92.83 |
| | Jun 25 | 1 | 68.26 | 64.35 | 96.93 | | Sept 3 | 1 | 68.92 | 66.48 | 85.14 |
| | | 2 | 79.49 | 82.065 | 98.19 | | | 2 | 71.72 | 69.555 | 86.15 |
| Sunflower | Aug 9 | 1 | 70.52 | 71.68 | 77.33 | Squash | Aug 9 | 1 | 69.61 | 53.595 | 96.32 |
| | | 2 | 76.75 | 75.12 | 82.79 | | | 2 | 71.05 | 80.075 | 98.86 |
| | Sept 3 | 1 | 68.52 | 65.32 | 74.74 | | Sept 3 | 1 | 68.69 | 63.285 | 96.99 |
| | | 2 | 70.32 | 73.75 | 79.69 | | | 2 | 71.31 | 87.61 | 99.27 |

The general models established by U-Net of the four crops in the key period could map the images at different times in the key period with high precision. Meanwhile, the constructed models were applied to the images in 2020 and had good results, indicating that the established mapping models had certain universality in different years. To explore the application of the proposed method to the multi-year mapping of large areas. The key periods and key bands of four crops in the HID were obtained from Table 4. The high-quality cloudless images in the key period in the HID from 2017 to 2021 were combined based on Table 1. Then, the constructed general mapping models were used to map the high-quality images, and the results were shown in Figure 8. The area was mainly mapped with six categories. Four categories of main crops were obtained through the general models. Cultivated land was obtained from the results in the literature [45]. Others were obtained by removing the vector map of five types of features from the vector map of the whole irrigation area in ArcGIS (10.2).

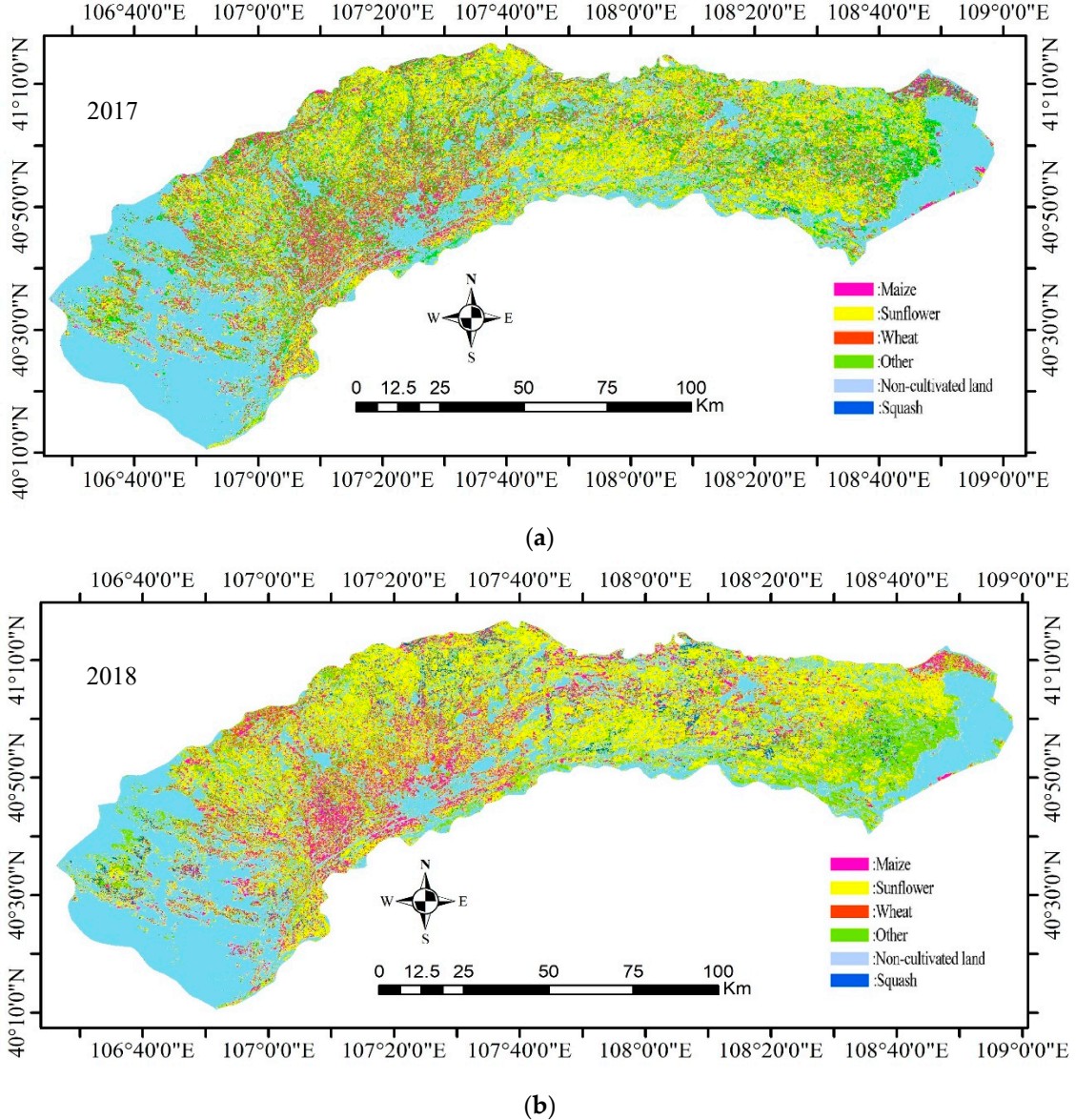

**Figure 8.** *Cont.*

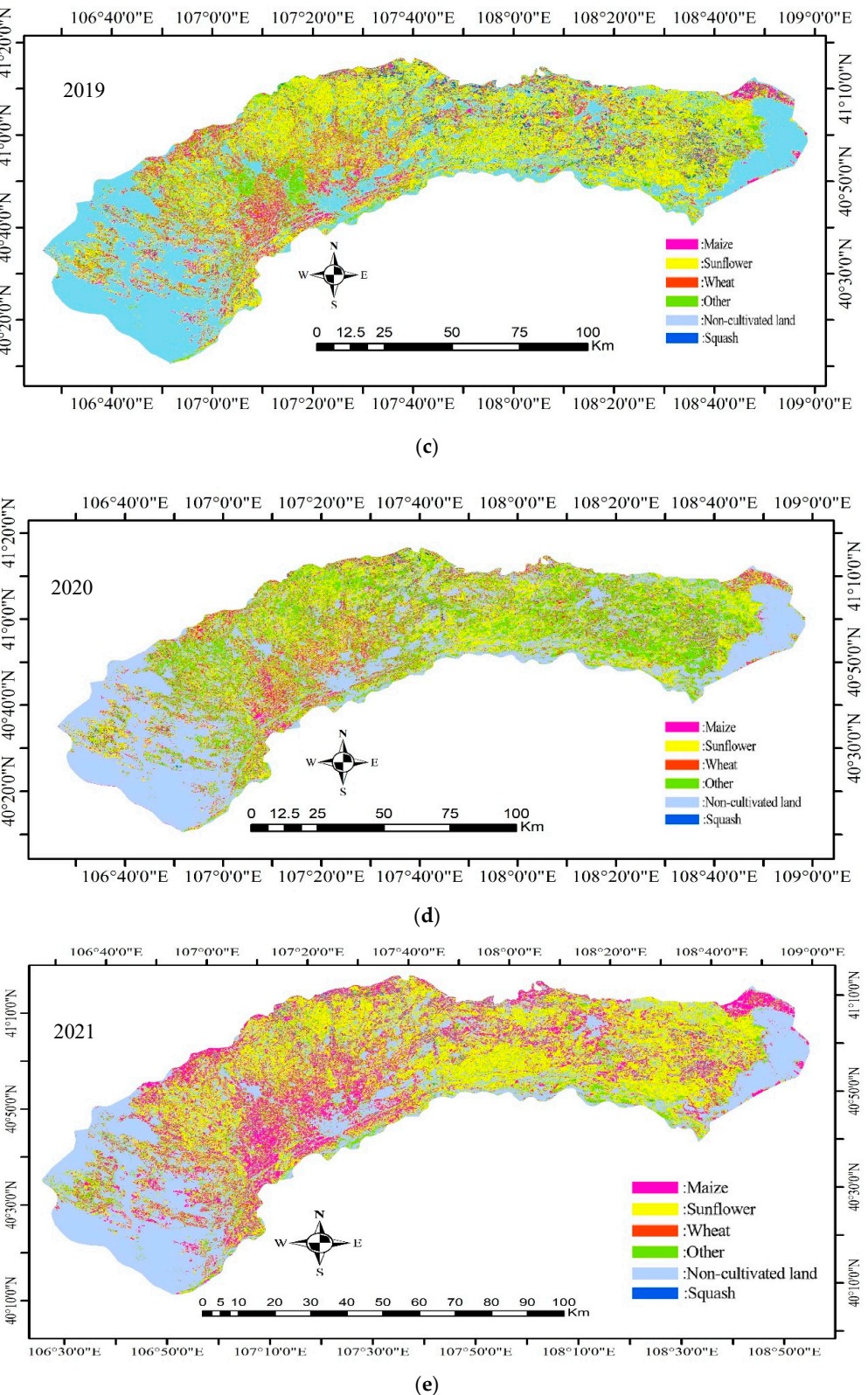

**Figure 8.** Mapping results of the four crops in the HID in different years. (**a**–**e**) represent the mapping results of 2017–2021, respectively. Different colors in the figure represent different types of objects in the HID. The pink areas represent the maize growing areas. The yellow areas represent the sunflower growing areas. The red areas represent the wheat growing areas. The blue areas represent the squash growing areas. The green areas represent other crops (such as tomato, green pepper, melon, etc.) in the cultivated land in the HID. The sky-blue area represents non-cultivated land in the HID.

## 4. Discussion

### 4.1. Advantages of Constructing Mapping Model in Key Period

Crop mapping in large areas is vulnerable to clouds and aerosols. Combining high-quality images at multiple adjacent time points is the main means to obtain high-quality images in large areas [46]; this can effectively avoid the pollution of clouds and aerosols. However, the images contain information of multiple times, which will enlarge the problem of "same object different spectrum, foreign object same spectrum". The GSI comprehensively considers the variability in the same crop and the differences in different crops, and comprehensively evaluates the separability of crops in the whole growth period. The periods with high separability of wheat, maize, sunflower, and squash in the HID are obtained, which is the time range of the combined images. Using all the images in the periods combined with the deep semantic segmentation algorithms, a general model suitable for the mapping of wheat, maize, sunflower, and squash in the HID in the key periods can be constructed. Compared with existing studies, the advantages are as follows. The existing research mainly uses algorithms and time series images to eliminate the influence of clouds. In terms of eliminating the influence of clouds by algorithms, existing research mainly uses the Q65 band to remove pixels with clouds in images, and then combine a deep learning framework or regression analysis model to reconstruct images, or use the percentile method to train models with time series images to avoid the influence of clouds [47–49]. These methods require a large number of data samples, as well as a series of complex calculations, and the reliability of the reconstructed data is difficult to guarantee. Compared with the existing methods, the multi-phase image combination method proposed in this paper is simpler and more directly eliminates the influence of clouds. Compared with the time series image mapping methods [19,49,50], it greatly and effectively reduces the demands of images and storage space and reduces the time and cost of data processing. Meanwhile, the average OA of the four crops in the HID can reach more than 88.28% by these methods, reaching the OA accuracy range of 85–95% for the current mapping using Sentinel-2 data [13,51] Therefore, synthesizing high-quality images of large areas combined with a general mapping model has a certain application value for large-area mapping.

### 4.2. Analysis of Adaptability of Different Algorithms to Sentinel-2 Mapping

Many studies in the literature have proved that the deep learning method has strong applicability to remote sensing mapping. A large number of deep semantic segmentation algorithms for remote sensing mapping have emerged [28,51]. Most of the research is aimed at high-resolution images, improving the algorithm to improve the mapping accuracy. Few studies have aimed at the applicability of models to imagery mapping in a period. Four representative deep semantic segmentation algorithms (U-Net, U-Net++, Deeplabv3+, and SegFormer) were selected to study the applicability of crop mapping in a time period. Compared with the other three algorithms, the mapping model constructed by U-Net has the higher accuracy. Figure 9 shown the prediction results of maize in sampling area 9 by the mapping model constructed by the four algorithms. Figure 10a–f shows the original image, label, and the prediction results of U-Net, U-Net++, Deeplabv3+, and SegFormer, respectively. From the yellow circle, the mapping result of U-Net has a clearer boundary than the other three methods, and can accurately identify the small fields and obtain better results. The reasons for the high mapping accuracy of the U-Net algorithm can be summarized into two aspects. Firstly, the highest spatial resolution of the images used for mapping is 10 m, which can provide limited ground feature detail information. Secondly, the four algorithms have different structures. The U-Net algorithim only carries out 4-layer downsampling and 4-layer upsampling and, thus, the structure is relatively simple. The backbone network structure of Deeplabv3+ and SegFormer is complex and carries out multi-layer downsampling and upsampling. Due to the limited monitoring ability of satellite images, even multi-layer downsampling cannot obtain the useful deep-level features of mapping but will affect the mapping accuracy. Meanwhile, we used Deeplabv3+ to combine the backbone networks with different depth layers (resnet18, resnet50, resnet101).

Using resnet18 obtained the best results, which shows that the structural complexity of the algorithm should be matched with the ability of image monitoring to obtain better mapping results.

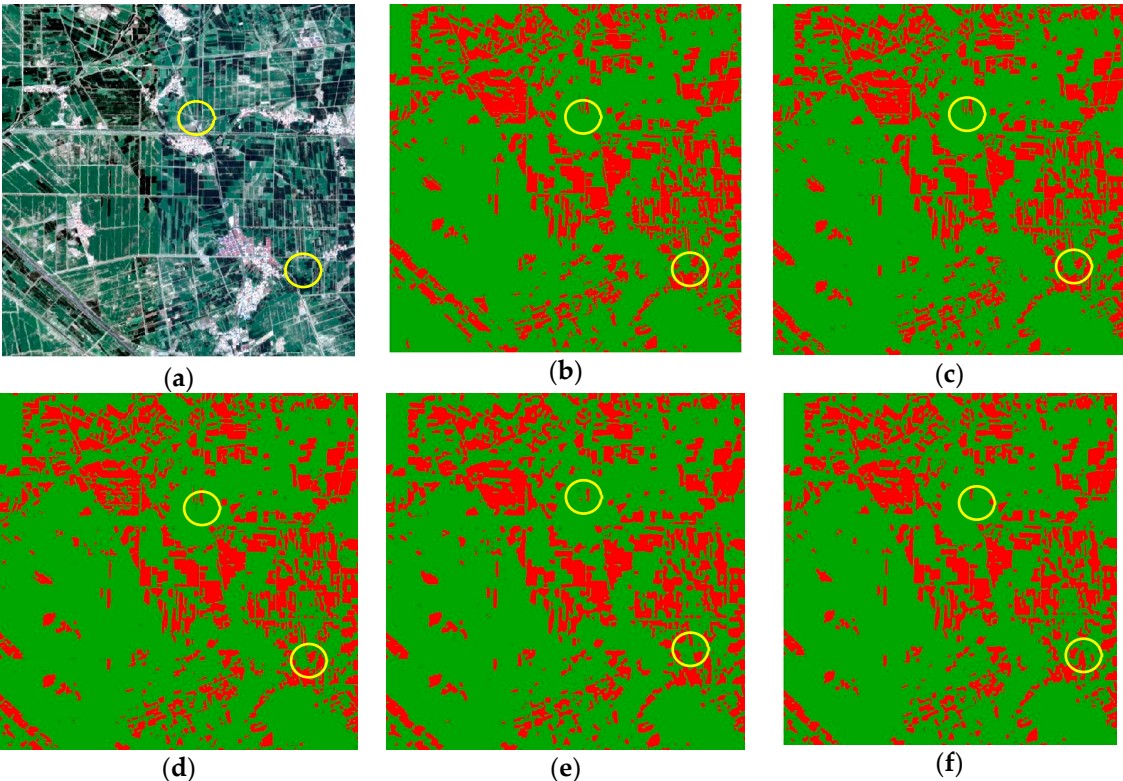

**Figure 9.** Mapping results of maize in the 2021 sample area by U-Net, U-Net++, Deeplabv3+, and SegFormer. (**a**) Original image, (**b**) label, (**c**) U-Net mapping result, (**d**) U-Net++ mapping result, (**e**) Deeplabv3+ mapping result, (**f**) SegFormer mapping result. Red represents the recognized maize, and green is the background. The yellow circles show the mapping differences between different algorithms for the same location.

*4.3. Applicability Analysis of the Model to Crop Mapping over Multiple Years*

The applicability of crop mapping models were mostly studied in different areas in the same year. Due to the lack of data in different years, few studies have discussed the applicability of crop type mapping models in different years. The high-quality images of the HID from 2017 to 2019 are obtained by image combination, and the mapping results of the HID from 2017 to 2019 are obtained by combining these with the constructed general mapping models. To explore its generality, the mapping results were compared with the crop planting statistics at the county level (http://www.nmgqq.com.cn/ accessed on 1 January 2022). Although the study area includes five counties, only Wuyuan County, Linhe District and Hanggin Rear Banner are fully included. Therefore, this part focuses on the data analysis of these three counties. County-level statistics can only reveal the area of grain crops, cash crops, and cultivated land. We combine the area of wheat and maize into grain crops, sunflower, squash, and other cash crops into cash crops, and all of the crops' area into cultivated land area. In Figure 10, we regression compared the remote sensing measurements of cultivated land area, cash crop area and grain crop area with the ground statistics in the three counties from 2017 to 2019. Figure 10a shown that the cultivated land area determined by remote sensing monitoring and ground survey was significantly correlated ($p < 0.01$). The fitting line was close to the line of 1:1, indicating that the remote sensing data had high availability. Figure 10b,c shown the relationship between the cash crop area and grain crop area monitored by remote sensing and ground

statistics. The fitting line of cash crops was higher than the 1:1 line, indicating that the area of cash crops obtained by remote sensing measurement was larger than the area of ground statistics. The fitting line of grain crops was lower than the 1:1 line, indicating that the area of grain crops obtained by remote sensing measurement was smaller than the area of ground statistics. The reasons include two aspects. Firstly, the planting scale of grain crops in the HID was small, and was distributed in small and broken plots. Due to the limited capacity of satellite monitoring, grain crops were easy classify as cash crops. Secondly, the grain crops estimated by satellite images only include maize and wheat. Rice and other grain crops were also distributed near the Yellow River in the south of the HID based on the actual investigation. They were separated into other crops and designated as cash crops, which caused the underestimation of grain crops and the overestimation of economic crops. Figure 10d comprehensively evaluated the extraction accuracy of grain, cash crops, and cultivated land area of the proposed method in the county area. To clarify the applicability of the proposed method to crop classification at the county level, we compared the classification accuracy of existing methods at the county level. In Figure 10d, the classification accuracy $R^2$ is 0.856, and the RMSE is 17,220.59 ha for the method proposed in this paper on the county level. Compared with the values of $R^2 = 0.75$ and RMSE = 39,698 ha in the literature [51], it has higher accuracy. As such, it shown that the crop mapping method in this paper has strong applicability over multiple years of data.

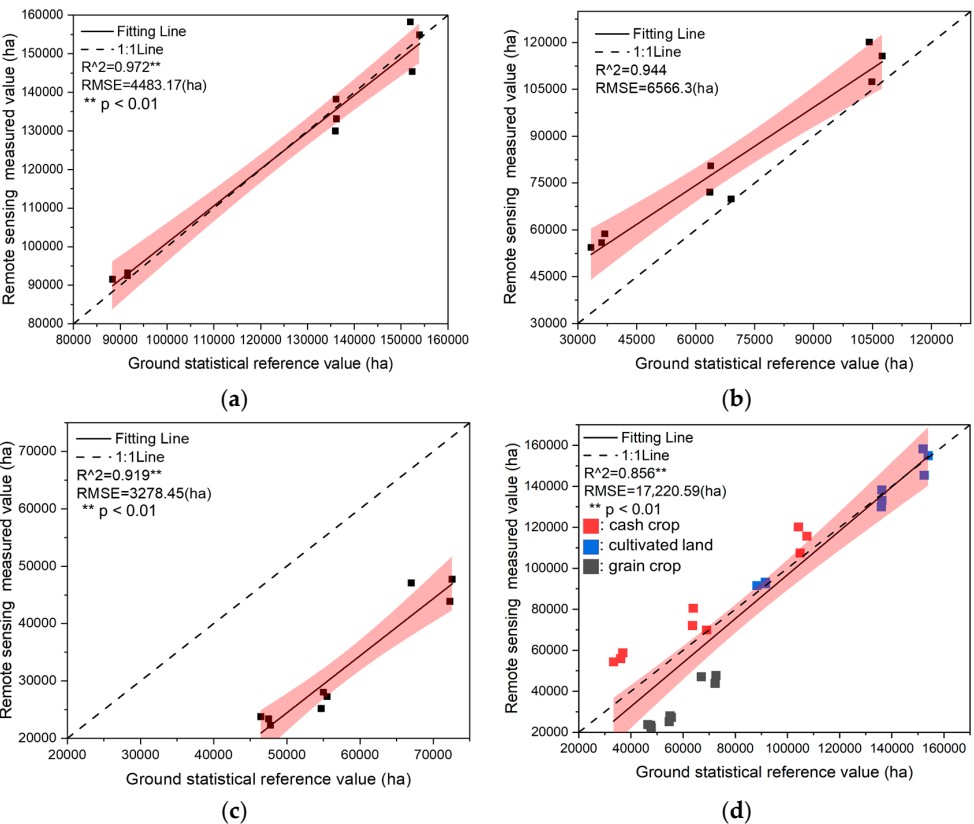

**Figure 10.** Regression comparison of cultivated land, cash crops, and grain crops at the county level based on remote sensing monitoring and ground survey. (**a**) Regression comparison of cultivated land area, (**b**) regression comparison of cash crop area, (**c**) regression comparison of grain crop area, (**d**) regression comparison of comprehensive data. Here, the county cultivated land area monitored by remote sensing was obtained by clipping the raster map of cultivated land in the literature [45] from the vector map of each county's administrative area. The cash crop area monitored by remote sensing refers to the total area of sunflowers, squash, and other crops monitored in each county. The grain crop area monitored by remote sensing refers to the total area of wheat and maize monitored in each county.

## 5. Conclusions

This paper studied the mapping methods of crop types over multiple years in large regions and drew the following conclusions. Constructing the GSI of crops in different periods can determine the key period of crop mapping. Combining multiple images in key periods can obtain high-quality images in large areas, which can effectively avoid the influence of clouds and aerosols. Using U-Net, U-Net++, Deeplabv3+, and SegFormer, combined with the multi-phase images in the key period, the general models in the key period are established, respectively. For mapping the test area in 2021, U-Net has higher mapping accuracy and is more suitable for mapping with Sentinel-2 imagery. The general model was applied to the mapping of the test area in 2020, and the average OA is more than 88.28%, which achieves the cartographic application accuracy of Sentinel-2 imagery. The mapping method is applied to the crop mapping in the HID from 2017 to 2019. By regression comparison of the obtained mapping area from remote sensing and the ground statistical area, the R2 is 0.856 and RMSE is 17,220.56 ha, indicating that the mapping method has a certain universality in different years.

**Author Contributions:** Conceptualization, G.L. and W.H.; data curation, G.L., S.H., W.M. and X.C.; formal analysis, G.L., Y.D. and X.Z.; investigation, G.L., S.H., W.M. and X.C.; writing—original draft, G.L.; writing—review and editing, W.H. and Y.W. All authors have read and agreed to the published version of the manuscript.

**Funding:** This study was supported by the National Natural Science Foundation of China [51979233], the Shaanxi Province Key Research and Development Projects [2022KW-47] and [2022NY-220], the Key Research and Development Project of Jiangsu Province (BE2021340), and the Natural Science Basic Research Program of Shaanxi Province [2022JQ-363].

**Data Availability Statement:** Not applicable.

**Acknowledgments:** We would like to thank Haipeng Chen, Jiandong Tang, Jianyi Ao, Chaoqun Li, and Jiawei Cui for their work on field data collection.

**Conflicts of Interest:** The authors declare no conflict of interest.

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
