# Peer review of "Multi-Year Crop Type Mapping Using Sentinel-2 Imagery and Deep Semantic Segmentation Algorithm in the Hetao Irrigation District in China"

_remotesensing, doi:10.3390/rs15040875_

Round 1
Reviewer 1 Report
This paper is well designed and provides interesting information for future applications.
Its orientation towards a real applicability in crops is very interesting.
The use of multispectral imaging Sentinel-2 is a widely used technology making it a very proven technology. The study has been designed correctly so the results are good.
Taking into account the above, I make the following comments with the intention of contributing as much as possible for the improvement of this paper. I think this could simply be done with minor corrections.

Reviewer 2 Report
This paper was designed to explore a universal model/method for crop classification mapping to address the current problems in crop mapping over large areas (i.e., low generality of models and susceptibility to cloud pollution). However, some previous studies have made similar investigation in Hetao Irrigation District (e.g., Hu, 2022, DOI: 10.3390/rs14051208). Please elaborate on the uniqueness of the work presented here. Other major and minor comments are below.
1. L101-108, relevant references should be cited here.
2. L140-141, there is an error in the description of the figures: Figure 1(a) shows the distribution of the sampling areas in 2021, Figure 1(b) shows the location of the study areas, and Figure 1(c) shows the distribution of the sampling areas in 2020. And in L182, I, II, III, IV shown in the purple box in Figure 1(c) should be moved to Figure 1(a).
3. In section 2.3.3, Figure 5 is almost the same as Wang et al., 2021(DOI: 10.11834/jrs.20210042), please cite the paper properly.
4. In section 2.4, the description about TN is wrong. True negative (TN): Observation is predicted negative and is actually negative. False negative (FN): Observation is predicted negative and is actually positive.
5. In section 3.2, the caption of Figure 7 says that the histogram with slashes represents the recognition results using all features, and the histogram without slashes represents the recognition results using the selected features. However, the legend says that histograms with slashes from the right to the left represents U-Net++, while those with slashes from the left to the right represents SegFormer, which is contradictory to the caption. Please reconcile and present the results of the four algorithms with the same features.
6. As field validation samples were absent from 2017 to 2019, have you considered comparing the results with existing mapping data to verify the universality of the model?
7. The format of some references is problematic: e.g., information on journal, volume and page numbers is missing for Reference 38 and 40.
Reviewer 3 Report
The authors proposed an improved method to map crop type using Sentinel-2 with deep learning algorithm. The results are quite interesting and provide useful information for the readers. The reviewer recommends following points to be revised.
1. How to combine multiphase images (L252)? More detail information is necessary.
2. The reviewer is interested in variation of GSI for each crop.
3. Does the term “features” mean spectral bands or another things?
4. Use larger font size in figures, and check the size in printed version. Some fonts can not be distinguishable.
5. Figure 1; the area of (c) should be shown in (a).
6. Figure 4; the color of labelled data should be consistent with Figure 9.
7. Figure 5; The image of segmentation output should be corrected. The image is the same as labelled data in Figure 4.
8. Figure 7 needs improvement. The figures are quite indistinguishable. The title should be “The highest and lowest results for mapping of crops using different features during key periods in 2021.” What “with/without slash” means?
9. The results in 2020 and 2021 are also shown in Figure 8.
10. Figure 11 (L493) seems Figure 10.
11. The definition of cultivated, cash and grain crops area for remote sensing measurement should be added in the caption in Figure 10. Different symbols should be used for each category in the comprehensive data in Figure 10.
Round 2
Reviewer 2 Report
I believe that the manuscript has been sufficiently improved to warrant publication in Remote Sensing.